# Toxicity Evaluation of Quantum Dots (ZnS and CdS) Singly and Combined in Zebrafish (*Danio rerio*)

**DOI:** 10.3390/ijerph17010232

**Published:** 2019-12-28

**Authors:** Beatriz Matos, Marta Martins, Antonio Cid Samamed, David Sousa, Isabel Ferreira, Mário S. Diniz

**Affiliations:** 1UCIBIO—Applied Molecular Biosciences Unit, Departamento de Química, Faculdade de Ciências e Tecnologia, Universidade NOVA de Lisboa, 2829-516 Caparica, Portugal; bi.matos@campus.fct.unl.pt (B.M.); ; marta.martins@fct.unl.pt (M.M.); acids@fct.unl.pt (A.C.S.); 2MARE—Marine and Environmental Sciences Centre, Departamento de Ciências e Engenharia do Ambiente, Faculdade de Ciências e Tecnologia, Universidade NOVA de Lisboa, 2829-516 Caparica, Portugal; 3LAQV/REQUIMTE—Laboratório Associado para a Química Verde, Departamento de Química, Faculdade de Ciências e Tecnologia, Universidade NOVA de Lisboa, 2829-516 Caparica, Portugal; 4CENIMAT/I3N—Centro de Investigação de Materiais /Institute for Nanostructures, Nanomodelling and Nanofabrication, Departamento de Ciência dos Materiais, Faculdade de Ciências e Tecnologia, Universidade NOVA de Lisboa, 2829-516 Caparica, Portugal; davidmagalhaessousa@gmail.com (D.S.); imf@fct.unl.pt (I.F.)

**Keywords:** quantum dots, CdS, ZnS, zebrafish, toxicity, oxidative stress

## Abstract

The exponential growth of nanotechnology has led to the production of large quantities of nanomaterials for numerous industrial, technological, agricultural, environmental, food and many other applications. However, this huge production has raised growing concerns about the adverse effects that the release of these nanomaterials may have on the environment and on living organisms. Regarding the effects of QDs on aquatic organisms, existing data is scarce and often contradictory. Thus, more information is needed to understand the mechanisms associated with the potential toxicity of these nanomaterials in the aquatic environment. The toxicity of QDs (ZnS and CdS) was evaluated in the freshwater fish *Danio rerio*. The fishes were exposed for seven days to different concentrations of QDs (10, 100 and 1000 µg/L) individually and combined. Oxidative stress enzymes (catalase, superoxide dismutase and glutathione *S*-transferase), lipid peroxidation, HSP70 and total ubiquitin were assessed. In general, results suggest low to moderate toxicity as shown by the increase in catalase activity and lipid peroxidation levels. The QDs (ZnS and CdS) appear to cause more adverse effects singly than when tested combined. However, LPO results suggest that exposure to CdS singly caused more oxidative stress in zebrafish than ZnS or when the two QDs were tested combined. Levels of Zn and Cd measured in fish tissues indicate that both elements were bioaccumulated by fish and the concentrations increased in tissues according to the concentrations tested. The increase in HSP70 measured in fish exposed to 100 µg ZnS-QDs/L may be associated with high levels of Zn determined in fish tissues. No significant changes were detected for total ubiquitin. More experiments should be performed to fully understand the effects of QDs exposure to aquatic biota.

## 1. Introduction

It is undeniable that humanity has entered an era of great development regarding nanotechnology. In fact, nanoscale manufacturing and the use of nanomaterials are already part of everyday life [1]. The progress made in this area in recent decades resulted in a tremendous impact on the industry with several uses in biomedicine, pharmaceutical, environment, electronics and many other fields [2,3]. Engineered nanomaterials (ENMs) (e.g., nanoparticles, carbon nanotubes, dendrimers, fullerenes, quantum dots) are usually developed using different types of materials (e.g., carbon, organic polymers, metals or metal oxides) and were defined as materials with sizes below 100 nm [4,5]. ENMs have different chemical and physical properties than those of their bulk counterparts as a result of high surface area to volume ratio and very small size [6]. Furthermore, ENMs can be manufactured with different shapes as tubular, spherical, flat, cubic or pillar [7,8].

Although the economic value of nanotechnology is well recognized, less attention has been given to the potential effects on ecosystems and particularly on the aquatic biota [9,10,11]. For example, much remains to be known about the fate and dispersion of ENMs in the environment and their main exposure routes [4,12,13]. According to a review by Libralato et al. [14], the increasing detection of ENMs in aquatic ecosystems is a direct consequence of the increased production and uses of these materials. Furthermore, developed models estimate that ENMs are present in surface waters ranging from ng/L to µg/L depending on the type of nanomaterial [15,16]. Nonetheless, the potential toxicological effects of ENMs to the aquatic biota are yet to clarify. Additionally, the possibility that ENMs can be bioaccumulated and transferred through trophic levels is also of great concern [17,18].

Quantum dots (QDs) are a distinct type of nanoparticles with several potential applications in the electronics industry and in biomedical applications [19,20]. QDs are semiconductor metalloid nanocrystals (~2–100 nm) showing unique optical and electrical properties [2,21,22]. Furthermore, zinc sulfide (ZnS), zinc–selenium (ZnSe), cadmium sulfide (CdS), cadmium–selenium (CdSe), and cadmium–tellurium (CdTe) cores belong to the group II-IV series QDs and are defined as particles with physical dimensions smaller than the exciton Bohr radius [23,24,25].

They are widely used as therapeutic agents and fluorescent dyes and are useful in the electronic field as LED displays, photovoltaic solar cells, ultrahigh-density data storage and quantum information processing [21,26,27]. The QDs core comprises diverse metal complexes (e.g., semiconductors, noble metals, and magnetic transition metals) and is usually encapsulated by a shell or a “cap” [19,21,28]. The shell improves QDs optical and electronic properties and reduces core metal leaching events, which are a major cause of toxicity [29,30]. The toxicity of QDs is variable as they can vary in their properties depending on the material of which they are made (e.g., core, capping) and other characteristics as size, surface chemistry or QDs concentration [14,23]. In addition, toxicity also depends on the living organism studied since different species may respond in a different way. Consequently, the current scientific literature reports conflicting and inconsistent results [14]. According to a review from Valizadeh et al. [19], QDs caused oxidative stress in plants, provoked toxicity in animal cells, but no effects were observed in amoeba exposed to QDs. Nonetheless, the toxicity caused by different types of QDs was reported in a great number of in vitro and in vivo studies as reviewed by Yong et al. [31]. Some of the major effects on cells and organisms are related to cell viability reduction, increased apoptosis and changes in the organism’s immune responses. Besides, several studies demonstrated that QDs are capable to generate reactive oxygen species (ROS) [32,33,34]. Therefore, the comprehension of the QDs potential toxicity is crucial and requires a fundamental understanding of QDs’ physical and chemical properties [21].

Zebrafish (*Danio rerio*) is one of the most commonly biological models used in developmental biology and molecular genetics. However, its great value for toxicological studies is now well established [35,36]. Due to their small size, easy maintenance, early morphology, reduced housing area, and low rearing costs, zebrafishes are now part of various risk assessment studies and programs [37].

Oxidative stress biomarkers have been widely used in nanotoxicology providing important information on the effects of exposure to nanomaterials and other environmental contaminants [38,39]. Generally, after exposure of organisms to ENMs, some biochemical and cellular responses may occur and indicate oxidative stress [40,41]. These responses activate the antioxidant defense system to protect the organism’s cells against the excessive production of ROS. Some of the antioxidant enzymes produced to fight oxidative stress are superoxide dismutase (SOD), catalase (CAT), glutathione peroxidase (GPX) or glutathione *S*-transferase (GSTs) which is an enzyme mainly involved in detoxification processes [11,14]. The impairment of this antioxidant system often results in a reduction of defense capacity and injury to cells by damaging membrane lipids, proteins, and DNA. Therefore, oxidative stress enzymes are a valuable tool to assess the toxic effects of QDs in exposed organisms [38,42,43]. However, much remains to be done regarding the risk assessment of QDs, especially for aquatic ecosystems and their biota, as they are the final repository of wastewater effluents.

Thus, the main objective of the present study is to assess the toxicity of different concentrations of QDs (CdS and ZnS), singly and combined, in the freshwater fish *Danio rerio*, by measuring the activities of antioxidant enzymes (superoxide dismutase, glutathione-S-transferase and catalase), lipid peroxidation, total antioxidant capacity, ubiquitin and HSP70. This work intends to provide additional information on the toxicity of QDs to fish, as increasing amounts are expected to be produced and used by industry in the coming years. Consequently, large quantities of QDs can be discharged to the aquatic ecosystems with an impact on the environment still unresolved and therefore more data on this subject is needed.

## 2. Materials and Methods

### 2.1. Quantum Dots (QDs) Synthesis

The QDs (CdS and ZnS) synthesis followed a method previously described by the authors [44], where zinc acetate dihydrate (ACS reagent, ≥98%) and cadmium acetate dihydrate (purum p.a., ≥98.0% (KT)), were added to 1-dodecanethiol (ACS reagent, ≥98%). The mixtures were purged with industrial-grade nitrogen and transferred to a Monowave 400 microwave reactor (Anton Paar, Graz, Austria) where they were heated to 300 °C in 10 min and kept at that temperature for 25 min. The suspensions were then centrifuged at 9000× rpm, and the pellet washed with ethanol 96%. This procedure was repeated four times. Afterwards, the resultant powders were washed using chloroform and centrifuged at 9000× rpm. The pellets were stored in the dark.

### 2.2. Preparation and Characterization of QDs (CdS and ZnS) Suspensions

The stock solutions of both QDs (1.0 g/L) were prepared using distilled water and then ultra-sonicated (10 min, 35 KHz at room temperature) using an ultrasonic bath (J-P Selecta Ultrasons HD, Barcelona, Spain). Subsequently, the stock solutions were stored in the dark at 4 °C until further use. To perform the bioassays, QDs suspensions were added to 10 exposure glass containers (1.0 L volume), containing previously filtered and de-chlorinated tap water, to obtain the following nominal concentrations: 10 µg CdS-QDs/L and 10 µg ZnS-QDs /L; 100 µg CdS-QDs /L and 100 µg ZnS-QDs/L; 1000 µg CdS-QDs/L and 1000 µg ZnS-QDs/L; 10 µg CdS-QDs /L plus 10 µg ZnS-QDs /L; 100 µg CdS-QDs /L plus 100 µg ZnS-QDs /L; 1000 µg CdS-QDs /L plus 1000 µg ZnS-QDs /L, which were selected based on models that estimate nanomaterial concentrations in aquatic systems [15,45].

The QDs were analyzed by scanning electron microscopy (SEM), transmission electron microscopy (TEM) and dynamic light scattering (DLS). For the SEM analysis, dispersions of all the QDs (CdS and ZnS) were applied on a carbon-coated adhesive, dried at room temperature and examined with a 1.00 to 2.00 keV field; scanning electron microscope (AURIGA microscope, Carl Zeiss, Oberkochen, Germany). For the TEM analysis, samples were dispersed in ethanol, dried on a Formvar and carbon coated 200 mesh copper grid and examined on a H-8100 II instrument (Hitachi, Tokyo, Japan) with a 200 kV electron beam at the MicroLab of the Instituto Superior Técnico (IST). The transformation of the QDs TEM images was performed with the aid of Gatan digital micrographs. Image J software was used to assess the size of the QDs by analyzing TEM images.

Dynamic Light Scattering (DLS) analyses were carried out in QDs suspensions (singly and combined) collected from the test glass containers, using a Nano particle Analyzer SZ-100 (HORIBA, Kyoto, Japan), with a Laser Diode (JUNO10G-HO, Showa. Optronics Co., Ltd., Yokohama, Japan) with an output of 10 mW (wavelength: 532 nm) and operating at 2 °C. Measurements obtained were processed using HORIBA SZ-100 software, furnished by HORIBA.

### 2.3. Bioassays

The test fish, *D. rerio* (*N* = 50 (× 2); 0.27 ± 0.15 g average weight; 2.74 ± 0.37 cm standard length), were obtained from commercial suppliers (Aquaplante, Lisbon, Portugal) and acclimated for one week in a closed-circuit system with filtered (activated carbon filtering) and de-chlorinated tap water. The fish were subjected to a photoperiod of 12h light to 12h darkness ratio, a temperature of 20 ± 1 °C and a pH of 7.4 ± 0.2, with continuous aeration enough for keeping the dissolved oxygen always higher than 6.0 mg/L. Then, zebrafish of both sexes, with less than one year of age, were randomly distributed by 10 glass containers, in groups of four fishes per container. The exposure assay was performed in duplicate. The glass containers had a capacity of 1.0 L volume of de-chlorinated water and the assay conditions (temperature, pH and dissolved oxygen) were the same as described before for the acclimation period. Some physical and chemical parameters as temperature (°C) (manual thermometer), ammonia (test kit API, Chalfont, PA, USA), pH and dissolved oxygen (Hanna Instrumentation, USA) were monitored during the assay. Fish were exposed for seven days to different concentrations of ZnS QDs and CdS QDs (10 µg/L, 100 µg/L and 1000 µg/L) singly and combined. Three containers containing filtered (carbon activated) and de-chlorinated tap water were used as controls. The experiment conditions in each container were renewed every 48 h. During the experiment period, fish were daily fed ad libitum (except 48 h before the end of the experiment) with commercial dry food (Eco vita - Anivite, Lisboa, Portugal) and the mortality rate was monitored daily. After the exposure period, fish were sampled and euthanized by freezing at - 80° for 5 min. Then fish were weighed and measured. Afterwards, the whole fish was homogenized as previously described [11,38]. In brief, fish were homogenized on ice-cold conditions, with the aid of a Tissue Homogenizer (Tissue Master 125, Omni, Kennesaw, GA, USA), in 3.0 mL of phosphate-buffered saline solution (PBS; 140 mM NaCl, (Panreac, Barcelona, Spain); 10 mM Na_2_HPO_4_, (Sigma-Aldrich, St. Louis, MO USA); 3 mM KCl, (Merck, Darmstadt, Germany); 2 mM KH_2_PO_4_, pH = 7.40, (Sigma-Aldrich). Tissue homogenates were transferred to microtubes (1.5 mL) and centrifuged 10,000×g (15 min at 4 °C) (VWR, model CT 15RE from Hitachi Koki Co., Ltd., Tokyo, Japan) and frozen at -80 °C until further analyses. All biochemical analyses were performed at least in duplicate. For normalizing results, the total protein mass (mg) was determined according to the Bradford (1976) method. A calibration curve was built using bovine serum albumin (BSA) standards (0 to 2.0 mg/mL).

### 2.4. Antioxidant Enzymes

#### 2.4.1. Catalase

The catalase (CAT) activity was determined as described by Johansson and Borg [46] following adaptation to 96-well microplates. This method is based on the reaction of the enzyme with methanol in the presence of an optimal concentration of H_2_O_2_. The formaldehyde produced is measured using 4-amino-3-hydrazino-5-mercapto-1,2,4-triazole (Purpald) as a chromogen. Briefly, 20 µL of each sample or standard, 100 µL of assay buffer (100 mM potassium phosphate) and 30 µL of methanol (Scharlab, Barcelona, Spain) were added to each well of a 96-well microplate (Greiner Bio-One GmbH, Frickenhausen, Germany). Then, the reaction was initiated by adding 20 µL of 0.035 M Hydrogen peroxide (30%) (Sigma-Aldrich) to all the wells and the microplate was incubated for 20 min on a shaker. Afterwards, 30 µL of 10 M potassium hydroxide (Chem-Lab, Zedelgem, Belgium) and 30 µL of 34.2 mM Purpald in 0.5M HCl (Sigma-Aldrich) were added into each well and incubated for 10 min in a shaker at room temperature. Afterwards, 10 µL of 65.2 mM Potassium periodate in 0.5 M KOH (Sigma-Aldrich, St. Louis, MO USA) was added to each well and allowed to incubate for 5 min. Then, the absorbance was read at 540 nm in a microplate reader (Synergy HTX, BioTek, Winooski, VT, USA). Formaldehyde concentration of the samples was determined based on a calibration curve using formaldehyde standards and prepared from a 4.25 mM formaldehyde (Sigma-Aldrich) stock solution to obtain a range of concentrations from 0 to 75 µM. Results are expressed in relation to the total protein concentration (nmol/min/mg).

#### 2.4.2. Superoxide Dismutase

The superoxide dismutase (SOD) assay was carried out using the nitroblue tetrazolium (NBT) method adapted from Sun et al. [47] for 96-well microplates. In this method, superoxide radicals (∙O_2_^−^) are generated by a reaction of xanthine with xanthine-oxidase (XOD) that reduces NBT to formazan. SOD competes with NBT for the dismutation of ∙O_2_^−^, inhibiting its reduction. In brief, 200 µL of 50 mM phosphate buffer (pH 8.0) (Sigma-Aldrich) was added to a 96-well microplate (Greiner, Bio-One GmbH, Frickenhausen, Germany), followed by adding 10 µL of 3 mM EDTA (Honey Well Riedel-de-Haën, Seelze, Germany), 10 µL of 3 mM xanthine (Sigma-Aldrich), 10 µL of 0.75 mM NBT (Sigma-Aldrich) and 10 µL of sample. Then, the reaction was initiated by adding 10 μL of XOD (Sigma-Aldrich) and the absorbance read at 560 nm, every 5 min, until reach 15 min, using a microplate reader (Synergy HTX, BioTek). The SOD results are expressed as a percentage (%) of inhibition of NBT-diformazan. For normalization purposes, the results were expressed in relation to the total protein concentration of the sample.

#### 2.4.3. Glutathione S-Transferase

The activity of glutathione-S-transferase (GST) was measured using a modified method originally described by Habig et al. [48] and optimized for 96-well microplates. In this assay, a conjugate of 1-chloro-2,4-dinitrobenzene (CDNB) and glutathione (GSH) is formed resulting in an increase in absorbance at 340 nm. The assay was performed by preparing a reaction mixture solution composed of 9.8 mL phosphate-buffered saline solution (PBS, pH 7.4), 100 µL of 200 mM L-glutathione reduced (Sigma-Aldrich) and 100 µL of 100 mM CDNB (Sigma-Aldrich). Then, 180 µL of this mixed solution plus 20 µL of sample or GST standard were added into each well of a 96-well microplate (Greiner). The absorbance was read at 340 nm at each minute for six minutes, using a microplate reader (Synergy HTX, BioTek). The change in absorbance per minute was then estimated and the reaction rate was determined using a CDNB extinction coefficient of 5.3 mM/cm. The results are expressed in relation to the total protein concentration of the sample.

### 2.5. Lipid Peroxides Assay (MDA Content)

The lipid peroxides assay followed the thiobarbituric acid reactive substances (TBARS) protocol as described by Uchiyama and Mihara [49]. Five μL of each sample was added to 45 μL of 50 mM monobasic sodium phosphate buffer. Next, 12.5 μL of sodium dodecyl sulfate (SDS 8.1%), 93.5 μL of trichloroacetic acid (TCA 20%, pH = 3.5) and 93.5 μL of thiobarbituric acid (TBA 1%) were added to each microtube. Then, 50.5 μL of MilliQ-grade ultrapure water was added into each microtube and agitated for 30s in a vortex. Subsequently, microtube lids were punctured with the aid of a needle and incubated in water (10 min. at 100 °C). Afterwards, they were placed on ice for a few minutes and allowed to cool, followed by adding 62.5 μL of MilliQ-grade ultrapure water and 312.5 μL of n-butanol pyridine (15:1, v/v) into each well. Then, microtubes were centrifuged at 5000× *g* for 5 min. Duplicates of 150 μL of the supernatant of each reaction were added into a 96-well microplate (Greiner) and absorbance was read at 530 nm (Synergy HTX, BioTek). The lipid peroxides were quantified by building a calibration curve (0–0.1 μM TBARS) using malondialdehyde bis(dimethyl acetal) (MDA) (Merck) as standard and, normalized dividing by the total amount of protein (mg).

### 2.6. Heat Shock Protein (HSP70) and Total Ubiquitin

Heat shock protein 70 (HSP70) and total ubiquitin were quantified through an indirect Enzyme-linked Immunosorbent Assay (ELISA) as previously described by Madeira et al. [50]. For HSP70 or total ubiquitin (TUb), 50 μL of each sample or respective standards were added to a 96-well microplate (Greiner Bio One) and incubated overnight at 4 °C. Then, the microplates were washed three times with a 0.05% PBS Tween-20 solution (Sigma-Aldrich) and 200 μL of blocking solution (PBS with 1% BSA) (Nzytech, Lisboa, Portugal) was added to each well and then incubated at 37 °C for 90 min (Labnet, Edison, NJ, USA). Afterwards, the microplates were washed again (PBS with 0.05% Tween-20) and the primary antibodies (HSP70 or TUb) were added to the respective microplates after proper dilution (0.5 μg/mL). Subsequently, for HSP70 50 μL of primary antibody (anti-HSP70/HSC70, Acris GmbH, Herford, Germany) or for total ubiquitin, 50 μL of primary antibody (Ub P4D1 (sc-8017, Santa Cruz, Dallas, TX, USA) were added into the microplate wells. After another washing step, 50 μL of secondary antibody (anti-mouse IgG Fc specific - alkaline phosphatase, Sigma-Aldrich; diluted to 1:1000 in 1% BSA) was added to each microplate well and incubated for 90 min. at 37 °C. Then, 100 μL of alkaline-phosphatase substrate (composed of 100 mM NaCl (Panreac, Barcelona, Spain), 100 mM Tris-HCl (Sigma-Aldrich), 50 mM MgCl_2_ (Sigma-Aldrich) and 27 mM PnPP (4-nitrophenyl phosphate disodium salt hexahydrate, pH 8.5, Sigma-Aldrich), were added to each microplate well and incubated for 30 min at room temperature. Finally, 50 μL of stop solution (3M NaOH, Panreac) was added to each microplate well and the absorbance measured at 405 nm, using a microplate reader (Synergy HTX, BioTek). For quantification purposes, a calibration curve was built using purified HSP70 active protein (OriGene, Rockville, MD, USA) to give a range of 0 to 2.0 μg/mL. Ubiquitin standards of purified ubiquitin (UBpBio, E-1100, Aurora, CO, USA), were prepared to use a concentration range of 0 to 0.8 µg/mL to construct a standard curve. The results for Hsp70 and total ubiquitin were expressed in relation to the total protein concentration (mg) of the sample.

### 2.7. Element Analysis (ICP-EAS)

The QDs concentrations were measured as total Zn and Cd in fish tissues and water samples by inductively coupled plasma atomic emission spectrometry (ICP-EAS) using a model Ultima apparatus (Horiba-Jobin Yvon, Kyoto, Japan). In brief, fish tissues (homogenates) were dried (70 °C), weighed, and then digested with 0.5 mL HNO_3_ (Merck) and by adding 10 µL of H_2_O_2_ (Sigma-Aldrich) plus 490 µL of H_2_O into each microtube. The water from bioassays was collected and samples were transferred to 1.5 mL microtubes. Then they were processed by adding 50 μL of HNO_3_ (Merck) into each microtube. Additionally, another assay was performed to investigate metal ion dissociation from QDs following the same experimental conditions described for the bioassays, except that no fishes or food was added to the experiment. After 48 h, water samples were collected from the test glass containers and then centrifuged in 2.0 mL microtubes (20,000× g for 60 min. at 4 °C). Then, samples were processed as previously described by Benavides et al. [11]. In Brief, after centrifugation, the supernatant was filtered through a 0.1-μm syringe filter and transferred to new microtubes (1.5 mL), followed by acid digestion with HNO_3_ (Merck) and H_2_O_2_ (Panreac). All samples were analyzed in triplicate. Analysis of certified biological material (ERM^®^-CE278k) was performed to validate the results. The total Cd and Zn (mg/Kg) measured in digested samples of reference material (Cd 0.336 ± 0.018; Zn 79.23 ± 3.95) were in accordance with the certified values (Cd 0.358 ± 0.025; Zn 71 ± 4).

### 2.8. Statistical Analysis

The statistical analysis of the results was carried out by using the non-parametric Mann-Whitney *U* test the statistical assumptions required for ANOVA were not accomplished. Statistical analyses were performed with a significance level of 5%, using the STATISTICA TM software 8.0 (StatSoft Inc., Palo Alto, CA, USA).

## 3. Results

Representative images from the transmission electron microscope (TEM) analysis of the QDs (CdS and ZnS), singly and combined in H_2_O, are presented in Figure 1. Macroscopic observation of the QDs showed that the CdS were orange in color, and the Zn had a pale pink appearance. The TEM images were analyzed with Image J software and showed an average size of 10 ± 2 nm and an irregular shape for ZnS QDs and an average size of 8 ± 2 nm and a spherical shape for CdS QDs. SEM analyses were also carried confirming the sizes of both QDs estimated from TEM images. Regarding DLS analysis, results are shown in Table 1. The results indicate that both QDs (ZnS and CdS) singly and combined show a trend to aggregate in water forming large aggregates. The Largest aggregates were found at the highest concentration (1000 μg/L) of QDs (ZnS and Cds) tested alone.

During the exposure period, no changes were observed for the water quality parameters (pH, temperature, and ammonia) monitored in the test glass containers, remaining at normal levels. Throughout the exposure period no significant mortality was observed (less than 10%).

The results for the different QDs (ZnS and CdS) concentrations tested (singly and combined) determined in the water samples by ICP-AES are presented in Table 2. The results show a great reduction of both QDs (singly and combined) in the water samples in comparison to the nominal concentrations. The results of metal ion dissociation from QDs are presented in Appendix A.

The results suggest that both Zn and Cd are present in the water samples as metal ions. However, a significant reduction was observed, varying from about 18 to 95%, depending on the tested dose and when compared to the total amount of Zn and Cd shown in Table 2.

The results from CAT, SOD, GST, and LPO in zebrafish exposed to the different QDs concentrations (singly and combined) are presented in Figure 2.

Regarding CAT specific activity (Figure 2a), a significant increase (*p* < 0.05) was found in fish exposed to 10 and 100 µg ZnS-QDs/L which showed the highest CAT average activities (17.42 ± 4.62 nmol/min/mg total protein). The lowest CAT average activities (6.76 ± 1.58 nmol/min/mg total protein) were measured in zebrafish exposed to 10 µg QDs/L (combined). Moreover, the statistical analysis revealed a significant increase (*p* < 0.05) in CAT activities in fish exposed to 100 and 1000 µg CdS-QDs/L.

The results from SOD expressed as percentage (%) of inhibition of NBT-diformazan (normalized to the total protein mass in the samples) in exposed zebrafish are shown in Figure 2b. No significant differences were found for all tested QDs concentrations (alone and combined). However, a general trend to increase of SOD was observed in accordance with the tested QDs concentrations (Figure 2b). The highest average SOD as percentage of inhibition (18.59 ± 5.02, % inhibition/ mg total protein) was found in fish exposed to 1000 µg/L CdS (QDs) (18.89 ± 3.55 % inhibition), whereas the lowest average levels (8.13 ± 3.40, % inhibition/ mg total protein) were measured in fish exposed to 10 µg QDs/L, combined.

With respect to GST activity (Figure 2c), no significant differences were found for all treatments in comparison to the controls. The highest GST average activities (43.10 ± 13.0 nmol/min/mg total protein) were measured in fish exposed to 100 µg ZnS-QDs/L and the lowest GST average activities (14.71 ± 2.91 nmol/min/mg total protein) were found in fish exposed to 1000 µg QDs/L combined. Additionally, it was registered a trend to increase in GST activities in fish exposed to both 10 and 100 µg/L QDs (ZnS and CdS) alone, followed by a trend to decrease in fish exposed to the highest concentration (1000 µg QDs/L) as shown in Figure 2c.

The results of lipid peroxidation analysis (MDA content) are presented in Figure 2d, showing the highest MDA average levels (17.11 ± 1.96 nmol/ mg of total protein) in fish exposed to 10 µg CdS-QDs/L, whereas the lowest average levels (6.90 ± 2.39 nmol /mg total protein) were detected in fish exposed to 10 µg QDs/L (combined). A significant increase was found in fish exposed to 10 µg CdS-QDs/L (*p* < 0.05) and in fish exposed to 1000 µg CdS-QDs/L (*p* < 0.05), in comparison to the respective controls (Figure 2d).

The results from HSP70 and total ubiquitin are presented in Figure 3. The HSP70 results show that the highest average levels of HSP70 (0.049 ± 0.025 μg/mg total protein) were determined in fish exposed to 100 µg/L ZnS (QDs). While the lowest average levels (0.0042 ± 0.0019 μg/mg total protein) were measured in fish exposed to 1000 µg ZnS-QDs/L.

The statistical analysis revealed a significant increase (*p* < 0.05) in HSP70 production which was measured in fish exposed to 100 µg ZnS-QDs/L (Figure 3a), whereas a significant decrease (*p* < 0.05) was found in the fish exposed to 1000 µg ZnS-QDs/L. With respect to total ubiquitin, the highest average levels (11.16 ± 5.08 ng/mg total protein) were detected in fish exposed to 100 µg ZnS-QDs/L, whereas the lowest average levels (6.05 ± 1.64 ng/mg total protein) were measured in fish exposed to 1000 µg QDs/L (combined). Additionally, the statistics revealed a significant increase (*p* < 0.05) in total ubiquitin in fish exposed to 100 µg QDs/L, while no significant differences (*p* > 0.05) were found for the other QDs tested concentrations, alone and combined.

The results for the different concentrations of QDs (ZnS and CdS) tested (alone and combined) determined in the whole fish tissues are presented in Table 3.

The elemental analysis (Cd and Zn) revealed that both elements accumulated in the whole fish tissues. Cadmium was below LOD in control fish. The statistics showed a significant increase (*p* < 0.05) in fish exposed to 10 and 100 µg ZnS-QDs/L, whereas a significant increase (*p* < 0.05) was found for al tested QDs concentrations in fish exposed to CdS and in fish exposed to the QDs combined.

## 4. Discussion

It is well known that aquatic ecosystems are particularly vulnerable to anthropogenic activities resulting in environmental pollution [51,52,53,54,55]. Regarding the behavior, fate and the effects of QDs on aquatic ecosystems, little is known. Furthermore, the great development of nanotechnology in recent decades led to the production of numerous new nanomaterials being discharged into the environment, especially to aquatic ecosystems [16,45,56].

With respect to the QDs used in the present work, the results from electron microscopy (TEM and SEM) confirmed an average size of ~5 to 10 nm for CdS-QDs and ~8 to 12nm for ZnS-QDs. This is important because particles with this size can easily enter organisms following exposure and be distributed through organisms’ tissues and cells causing injury. In fact, other studies demonstrated this capability of QDs to enter organisms’ cells and cause damage to several organisms as mice [57,58], fish [34,59], *Daphnia* [60,61], mussels [62,63] or cell lines (e.g., see for a review [43]) among others species.

Nonetheless, the DLS results from the present study suggest that the QDs tended to form large aggregates in water suspensions. The tendency of ENMs (e.g., nanoparticles, quantum dots, nanotubes) to aggregate in aqueous suspensions was also reported by other authors [4,64,65]. On the other hand, the ICP-AES results from the water samples show a reduction in the actual concentrations of QDs in the water compared to the nominal concentrations, which was higher at the highest concentrations tested (a reduction of more than 90% depending on the assay). A possible explanation is that this reduction is linked to the QDs aggregation, as shown by the DLS and ICP-AES analyzes, and thus lower amounts are dispersed in the medium. This is in agreement with other studies reporting that QDs in water tend to aggregate to some extent, depending on several physical and chemical factors [60,66,67,68]. In fact, the ENMs fate and behavior in the aquatic environment are dependent on several factors (e.g., size, shape, charge, coating and chemical composition) and the presence of organic matter which will influence particle aggregation [69]. We can also hypothesize that some Cd and Zn analyzed in the samples (water and tissues) were metal ions released from the QDs. The results from the metal ion dissociation from the QDs core may explain some effects observed in fish, suggesting that there is some degree of metal ion release which is contributing to the sub-lethal effects observed for some biomarkers analyzed. In fact, previous studies showed that metal NPs ion dissociation was dependent on size decreasing with larger aggregates [11,66], which is compatible with the results from the present study.

Indeed, other authors have attributed observed toxicity to metal ions leaching from the QDs core [34,70,71,72,73]. Similar results were also reported by Benavides et al. [11] after exposing *C. auratus* to AlO_3_ and ZnO nanoparticles (NPs). It must be highlighted that surface charge plays also an important role in ENMS toxicity as it decisively influences interactions with biological systems. According to a review from Gatoo et al. [7], positively charged NPs show higher cellular internalization compared to negatively charged NPs. On the other hand, surface charge alters ENMs shape and size by forming aggregates or agglomerates [74].

Regarding the water contaminated with nominal 100 µg Zn-QDs/L, ICP-AES results showed levels of total Zn greater than expected. The determined total Zn levels were above the nominal concentration only for the Zn-QDs tested alone. This higher value must be interpreted with caution. Nonetheless, we can attribute this result to excess of aggregates as upon sample collection a few more aggregates can be collected influencing the total amount of Zn in the sample. In fact, if some aggregates are present in the water samples, then the ICP-AES measurements can overestimate the amount of the total element in the sample. Moreover, this suggests that the use of other quantitative techniques (e.g., ICP-MS, XRF) could be useful to complement the QDs analysis and provide more accuracy on results. In addition, Mourdikoudis et al. [75] stated that there are significant challenges in the analysis of ENMs due to the interdisciplinary nature of the field, the lack of suitable reference materials for the calibration of analytical tools, the difficulties associated to the sample preparation and the interpretation of the data.

In the present study, the results show that no significant mortality occurred during the exposure period. This means that none of the different QDs concentrations tested were enough to cause death in fish. The results from ICP-AES analyzes in tissues showed that fish accumulated both elements in accordance with the increasing concentrations of QDs tested even though a great reduction of the nominal concentrations was observed. However, the results from the biochemical analyses suggest that fish exposed to 10 and 100 µg QDs/L revealed more pronounced adverse effects.

SOD and CAT are the first enzymes acting sequentially by transforming ROS into H_2_O_2_ and then into H_2_O and O_2_ [11,76,77]. In general, the biochemical results showed low or moderate sub-lethal effects in exposed fish. Although there was a tendency for SOD to increase according to the concentrations of QDs tested no significant differences were found. However, CAT activities showed an overall increase in fish exposed to ZnS and CdS, suggesting that CAT is counteracting H_2_O_2_ overproduction. GST activities in exposed fish showed a similar trend to that observed for CAT, however, fish exposed to QDs combined (ZnS + CdS) showed decreasing GST activities compared to controls. For example, although there are some contradictory results, a study by Sadicck et al. [78] showed that CAT and GST activities increase after exposure of freshwater fish (*Oreochromis niloticus* and *Tilapia zillii*) to ZnNPs.

Since GST can fight oxidative stress, we can hypothesize that GST is also fighting against oxidative stress induced by exposure to QDs. In fact, living organisms have defense mechanisms to counteract the ROS overproduction, usually a set of enzymes (e.g., CAT, SOD, GPx) but also other compounds as tocopherols, carotenes, vitamin A, and ubiquinols. When the capacity of the antioxidant system is exceeded then negative effects on the organisms arise [76,77].

The MDA content in an organism is usually used as an indicator of lipid peroxidation due to oxidative stress in cells [11,79]. The results from LPO showed variable values for the different QDs concentrations tested. However, an increase was noticed in fish exposed to the different concentrations of CdS suggesting that it caused more pronounced effects in the cell’s membranes than ZnS. Interestingly, in this work when QDs were tested combined, lipid peroxidation decreased. Thus, higher QDs concentrations produced lesser toxicity than lower concentrations, which can be partially attributed to QDs aggregation. Furthermore, Gatoo et al. [7] stated that with an increase in the concentration of nanoparticles, the toxicity decreases and Lewinski et al. [27] reported that higher QDs concentrations caused less toxicity in zebrafish because after ingestion they suffer minor chemical degradation due to the absence of a stomach and no acid phase digestion in this species. Still, the higher results for antioxidant enzymes in fish exposed to 100 µg QDs/L can be explained by the action of the antioxidant system to counteract the oxidative stress caused by the exposure to QDs.

The HSP70 production in *D. rerio* exposed to the different QDs concentrations (tested singly or combined) showed an increase in fish exposed to 100 µg ZnS-QDs/L. These results are compatible with CAT, SOD and GST results. However, caution is advised when interpreting the results since the increase in GST and CAT activities can be associated with the unexpected elevated levels of Zn in fish exposed to nominal 100 µg ZnS-QDs/L. Similarly, high zinc levels may be related to the increase in HSP70 observed at this exposure concentration. In fact, several studies showed that Zn induces the production of HSP70 to protect cells [80,81].

Gagné et al. [82]) exposed primary cultures of rainbow trout hepatocytes to CdTe-QDs showing induction of HSP70 and suggested that the cytotoxic response was due to Cd ions release. It is known that beyond HSP70 induction during thermal stress, HSP70 acts also as a chaperone to protect cells functioning [83,84]. Thus, the rise of HSP70 detected in fish exposed to 100 µg QDs/L can be attributed to a biochemical response to protect cells. In addition, heat shock proteins were associated with membrane lipids, preserving cell membrane structure and integrity during the early stages of stress conditions, regulating membrane fluidity [85] and preventing cell death by stabilizing the lysosomal membrane [86].

Concerning total ubiquitin, the results were variable, and no significant differences were found in comparison to the respective controls. Since ubiquitin targets damaged proteins to be degraded in the proteasome preventing cytotoxicity [87], this can suggest that the antioxidant defense system is acting to protect cells.

Additionally, the high variability found in the controls for some biomarkers (e.g., SOD, ubiquitin) can be attributed to factors such as individual variability, sex, genetic heritage or other biological and environmental factors to which the animal has been subjected [88]. Nonetheless, the statistics showed no significant differences between each group of controls.

Overall, it can be stated that after zebrafish exposure to QDs the cellular system fought oxidative stress and prevented the development of marked adverse effects. The fact that QDs are functionalized with 1-dodecanethiol (C_12_H_25_SH) and aggregated in the medium also contributed to the reduced toxicity.

The current scientific literature shows apparently contradictory results. For instance, there are studies showing low or no significant toxicity: Blickley et al. [34] fed estuarine fish (*Fundulus heteroclitus*) with one or 10 μg of lecithin-encapsulated CdSe/ZnS QD/day and no significant changes in hepatic total glutathione, lipid peroxidation or oxidative stress was observed, Guo et al. [89] exposed zebrafish embryo to graphene QDs (12.5–200 μg/mL for up to 96h and no significant toxicity was observed at concentrations below 50 μg/mL, Galdiero et al. [90] showed that QDs functionalized with indolicidin caused low toxicity to *D. magna*, Lewinski et al. [27] studied the trophic transfer of amphiphilic polymer-coated CdSe/ZnS QDs by feeding fish (10 QD contaminated Daphnia) to *D. rerio*, but no significant adverse effects were noticed. On the other hand, several studies showed that QDs caused different degrees of toxicity in vivo and *in vitro*. Though, it depends on the physical and chemical properties of the QDs, capping, dose, model organisms and exposure time (e.g., for a review see [14,21,22]).

Furthermore, although some studies indicate low toxicity for short-term QDs exposure, other studies point increased toxicity when longer exposure periods are considered, as shown by Aye et al. [91], who studied the long-term in vitro toxicity effects of lipid-coated CdSe/ZnS QDs or Galeone et al. [92] that found long-term toxicity of CdSe-ZnS QDs in *Drosophila melanogaster*.

In general, the low or moderate sub-lethal effects found in the present study can be attributed to multiple factors as QDs aggregation, functionalized group, dose, physical and chemical characteristics of QDs and the organism’s capability to counteract oxidative stress. Nonetheless, since studies report conflicting results more studies should be conducted to a better comprehension of the QDs effects on the aquatic biota.

## 5. Conclusions

The short-term exposure of adult *D. rerio* to different QDs concentrations (ZnS and CdS), singly and combined, did not cause significant mortality in fish. In addition, the exposure to QDs caused a low to moderate stress oxidative response, as shown mainly by the CAT and LPO responses. The results suggest that ZnS and CdS tested alone caused more pronounced effects than when combined. In addition, LPO results suggest that CdS caused more damage to cell’s membrane than ZnS or when QDs were tested combined. It was observed a great reduction of QDs (measured as total Zn and Cd) concentrations in the water from the bioassays. Nonetheless, total Zn and Cd measured in whole fish tissues show bioaccumulation according to increasing concentrations. Additionally, more studies using different species, testing other types of QDs, including relevant concentrations and mixtures are required. This is essential to clarify the possible effects and mechanisms involved in the toxicity of QDs to aquatic organisms. Moreover, further studies need to be conducted to understand what happens to this type of nanomaterials in the environment. That is, what changes occur at the physical and chemical level and what are the implications in terms of toxicity.

## Figures and Tables

**Figure 1 ijerph-17-00232-f001:**
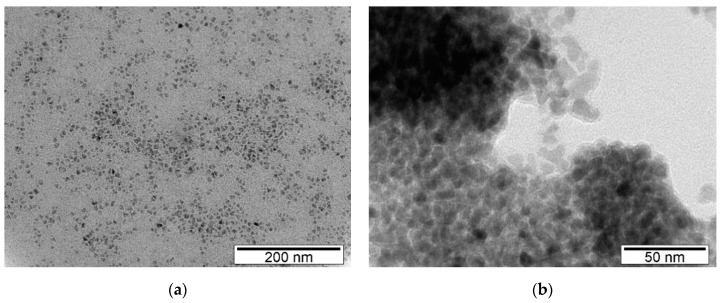
Representative TEM images of QDs: (**a**) ZnS and (**b**) CdS.

**Figure 2 ijerph-17-00232-f002:**
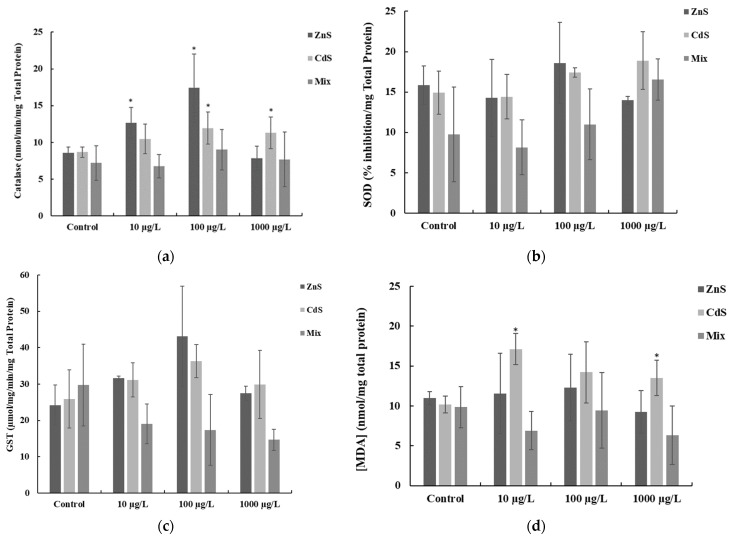
Antioxidant enzymes activities (mean ± SD): (**a**) CAT, (**b**) SOD (**c**) GST; and (**d**) Lipid peroxidation (MDA content) in zebrafish exposed to different concentrations of QDs, alone and combined (Mix). Asterisk means significant differences (*p* < 0.05) in comparison to the respective controls.

**Figure 3 ijerph-17-00232-f003:**
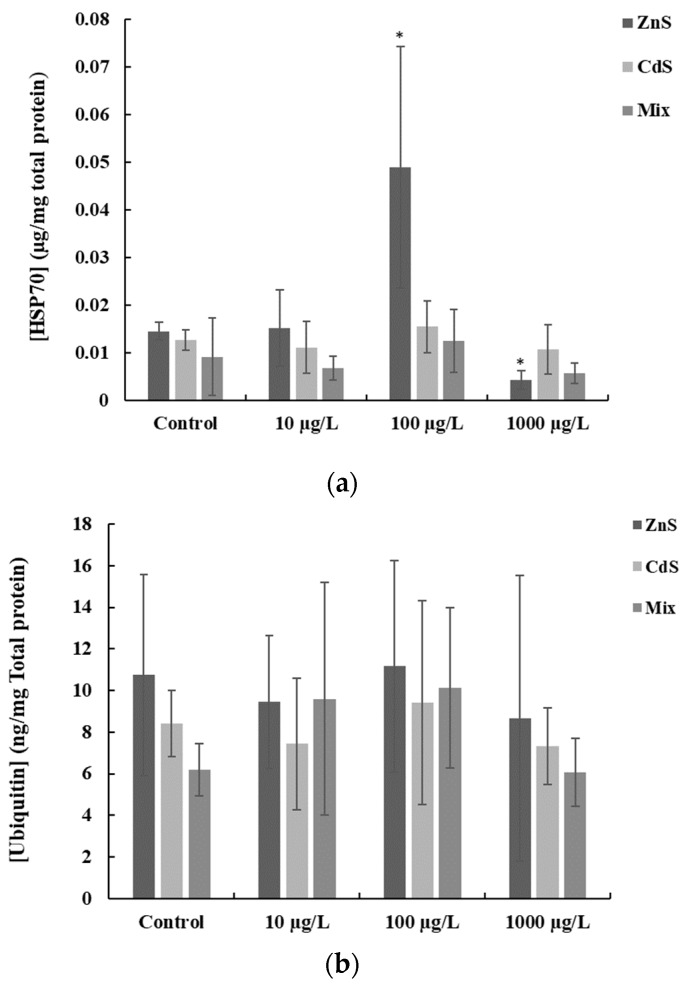
(**a**) HSP70 and (**b**) total ubiquitin (mean ± SD) in zebrafish exposed to different concentrations of QDs, alone and combined (Mix). Asterisk means significant differences (*p* < 0.05) from the respective controls.

**Table 1 ijerph-17-00232-t001:** Results from DLS analysis (mean ± sd), potential zeta and electrophoretic mobility in water samples.

Quantum Dot	Mean Size ± sd	Potential Zeta	Electrophoretic Mobility Mean (cm^2^/V)
ZnS 10	289 ± 18 nm (PDI 1.027)	−26.8 mV	−0.000208
ZnS 100	511 ± 163 nm (PDI 0.938)	−15.5 mV	−0.000120
ZnS 1000	1062 ± 363 nm (PDI 0.804)	−60.8 mV	−0.000316
CdS10	566 ± 61 nm (PDI 1.268)	−40.8 mV	−0.000316
CdS100	416 ± 55 nm (PDI 1.784)	−24.2 mV	−0.000188
CdS 1000	4671 ± 825 nm (PDI 0.557)	−76.7 mV	−0.000397
ZnS + CdS 10	658 ± 310 nm (PDI 1.145)	−24.5 mV	−0.000191
ZnS + CdS 100	770 ± 108 nm (PDI 0.467)	−22.8 mV	−0.000770
ZnS + CdS 1000	596 ± 119 nm (PDI 0.013)	−42.9 mV	−0.000222

PDI: polydispersity index.

**Table 2 ijerph-17-00232-t002:** Results from ICP-AES analysis of Zn and Cd in water samples.

Element Analysed (μg/L)		
Assay	Zn	Cd
Control	2.88 ± 0.14	<LOQ
10 µg ZnS/L	5.93 ± 0.36	-
100 µg ZnS/L	123.96 ± 12.39	-
1000 µg ZnS/L	43.02 ± 4.15	-
10 µg CdS/L	-	7.05 ± 0.57
100 µg CdS/L	-	50.22 ± 9.87
1000 µg CdS/L	-	71.31 ± 15.22
10 µg (ZnS + CdS)/L	4.17 ± 0.31	9.24 ± 1.46
100 µg (ZnS + CdS)/L	31.45 ± 3.20	40.04 ± 6.11
1000 µg (ZnS + CdS)/L	50.69 ± 8.54	74.85 ± 11.30

Significant differences from controls if (*). LOD: Cd (0.6 μg/L); Zn (0.3 μg/L). LOQ: Cd (2.0 μg/L); Zn (1.0 μg/L). *n* = 3.

**Table 3 ijerph-17-00232-t003:** Results from ICP-AES analysis of Zn and Cd in whole fish tissues exposed to the different concentrations of QDs (singly and combined).

Element Analysed (g/Kg Dry Weight)		
Assay	Zn	Cd
Control (ZnS)	55.98 ± 9.42	-
10 µg ZnS/L	76.0 ± 22.31	-
100 µg ZnS/L	105.81 ± 26.63 *	-
1000 µg ZnS/L	106.58 ± 13.14 *	-
Control (CdS)	-	<LOD
10 µg CdS/L	-	0.05 ± 0.02 *
100 µg CdS/L	-	0.86 ± 0.52 *
1000 µg CdS/L	-	3.14 ± 1.45 *
Control (ZnS + CdS)	293.0 ± 8.14	<LOD
10 µg (ZnS + CdS)/L	311.32 ± 15.41	0.14 ± 0.02 *
100 µg (ZnS + CdS)/L	292.41 ± 33.68	1.62 ± 0.77 *
1000 µg (ZnS + CdS)/L	289.32 ± 19.91	5.28 ± 2.72 *

Significant differences from controls if (*). LOD: Cd (0.6 μg/L); Zn (0.3 μg/L). LOQ: Cd (2.0 μg/L); Zn (1.0 μg/L). *n* = 4.

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
