# Peer review of "Toxicity Evaluation of Quantum Dots (ZnS and CdS) Singly and Combined in Zebrafish (Danio rerio)"

_ijerph, 2019, doi:10.3390/ijerph17010232_

Round 1
Reviewer 1 Report
The authors expose adult zebrafish to three different concentration Zn and Cd quantum dots. The authors assayed markers associated with oxidative stress, lipid peroxidation, HSP7- and total ubiquitin. Some affect is identified in these fish.
How were concentrations choosen and how do these concentrations compare to previous exposures to quantum dots of the same composition? Authors should justify why these concentrations are used. Would any these concentrations be relevant in an environmental exposure situation. There is no presented evidence that these quantum dots concentrate with increasing trophic layers of the food chain.
In table 2 it appears that there is a higher concentration of Zn in the water column than the QDs that are put into the solution. 100 ug/L are put in and there is 123.96 detected. Authors should explain.
As the authors mention much of the QDs aggregate and presumably settle to the bottom of the container. Was there any attempt to keep QDs within the water column during the exposure?
The authors do not specify the level of soluble metal released by the QDs into the water column. The metal concentration released by the QDs may result in the observed changes to the fish. Authors sample the water column but do not remove QDs. Authors should analyze level of solubilized metal following removal of QDs. If the levels of soluble metal are relevant, the authors should expose fish to similar concentration of metal to determine whether soluble metal will have the same effects as the QD exposure. There may at least be a contribution of soluble metal to the observed changes.
Author Response
Reviewer 1
Comments and Suggestions for Authors
The authors expose adult zebrafish to three different concentration Zn and Cd quantum dots. The authors assayed markers associated with oxidative stress, lipid peroxidation, HSP7- and total ubiquitin. Some affect is identified in these fish.
How were concentrations choosen and how do these concentrations compare to previous exposures to quantum dots of the same composition? Authors should justify why these concentrations are used. Would any these concentrations be relevant in an environmental exposure situation. There is no presented evidence that these quantum dots concentrate with increasing trophic layers of the food chain.
R1: Currently there is very few information on environmental concentrations of QDs with the same composition. Nonetheless, in the Discussion section, we used the most relevant scientific literature regarding QDs near in composition with the ones used in the present study.
We selected a range between 0 to 1000 mg/L, since we think that might represent relevant to non-relevant environmental concentrations, based on nanomaterials concentrations estimated by some authors and expected to be found in the aquatic systems. We referred this in the manuscript (15,16) Moreover, the higher concentrations also allow a better understanding of the QDs effects in fish. We added more information for the selected concentrations to the M &M section.
The aim of the present study was not to assess trophic transfer. The goal was focused on biomarkers of effect and not to evaluate trophic transfer.
In table 2 it appears that there is a higher concentration of Zn in the water column than the QDs that are put into the solution. 100 ug/L are put in and there is 123.96 detected. Authors should explain.
R1: This increase of more 24mg/L than expected can be attributed to unknown contamination or possibly to the contribution of some Zn present in food residues that were not ingested by fish. It must be noticed that fish exposed to 100mg/L had a different behavior (possibly due to the QDS exposure) and different from the other tested concentrations they tend not to ingest the same amount of food than the other fish. Thus, more residues of food were present in the water column possibly contributing with the Zn present in food pellets. We added an additional comment on this issue in the manuscript.
As the authors mention much of the QDs aggregate and presumably settle to the bottom of the container. Was there any attempt to keep QDs within the water column during the exposure?
R1: Yes, as reported in many studies the aeration should be strong enough to mix and keep within the water column. However, some of the QDs may aggregate with organic matter besides aggregation with itself and sediment at the bottom anyway. This phenomenon is often described by other studies (Petosa et al. 2010).
The authors do not specify the level of soluble metal released by the QDs into the water column. The metal concentration released by the QDs may result in the observed changes to the fish. Authors sample the water column but do not remove QDs. Authors should analyze level of solubilized metal following removal of QDs. If the levels of soluble metal are relevant, the authors should expose fish to similar concentration of metal to determine whether soluble metal will have the same effects as the QD exposure. There may at least be a contribution of soluble metal to the observed changes.
R: We agree with the reviewer and we performed an additional test to quantify the ionic contribution (metal ions dissolved in the water samples). The results were added to the manuscript as supplementary material (table SM1). Regarding an additional assay to test soluble metal effects in fish, it is not feasible within the time of manuscript revisions (10 days), however, as shown in the Discussion section, we compared with other studies that correlated the results with metal ion release from QDs core (References: 7, 34, 70-73).
Reviewer 2 Report
Manuscript Number: ijerph-643813
Title: Toxicity evaluation of QDs (ZnS and CdS) singly and combined in zebrafish (Danio rerio)
The manuscript deals with the exposure related toxicity of QDs (ZnS and CdS), in 1 year adult zebrafish. Animals were exposed to QDs for a relatively short period of time (7 days) under laboratory conditions, and activities of antioxidant enzymes were assessed and discussed.
The paper is extremely well prepared and the amount of work behind it is impressive; however it requires some minor corrections and clarifications prior to acceptance for publication.
Please write the full name (quantum dots) in the title Introduction, Line 70: "In addition, the toxicity depends on the living....." Materials and Methods: Not sure if the methods for Quantum Dots synthesis and the fish homogenisation are adapted from another paper, or originally developed by the authors. Please specify. The experimental conditions (temperature, pH etc) seem to be identical with the acclimatisation ones. Therefore Line 146: the sentence should just acknowledge the identical conditions. The results should try to explain or even acknowledge the high variability of data in the control samples, especially for SOD and Ubiquitin (Fig 2 and 3). Conclusions. Line 445: "The short exposure of adult..."The authors should discuss another factor, potentially responsible for the low or moderate effects found in this present study - the short exposure. I recommend Nesler et al, 2019 (Toxicology in vitro) - the potential of high impact after long exposure compared to low exposure is highlighted here.
Author Response
Reviewers' Comments to Authors and authors answers:
Reviewer 2
Comments and Suggestions for Authors
Manuscript Number: ijerph-643813
Title: Toxicity evaluation of QDs (ZnS and CdS) singly and combined in zebrafish (Danio rerio)
The manuscript deals with the exposure related toxicity of QDs (ZnS and CdS), in 1 year adult zebrafish. Animals were exposed to QDs for a relatively short period of time (7 days) under laboratory conditions, and activities of antioxidant enzymes were assessed and discussed.
The paper is extremely well prepared and the amount of work behind it is impressive; however it requires some minor corrections and clarifications prior to acceptance for publication.
Please write the full name (quantum dots) in the title Introduction
R2: done
Line 70: "In addition, the toxicity depends on the living....." Materials and Methods: Not sure if the methods for Quantum Dots synthesis and the fish homogenisation are adapted from another paper, or originally developed by the authors. Please specify.
R2: Yes, the QDs synthesis was performed by the authors by adapting a technique previously described in a previous paper from the authors: Sousa et al. (2018) Facile Microwave-assisted Synthesis Manganese Doped Zinc Sulfide Nanoparticles. Sci. Reports.
This reference was added to the manuscript and the sentence was rewritten.
Fish homogenization is a routine procedure using tissue grinders or electric homogenizers to promote cell lysis and obtain the cytosolic fraction. Nonetheless, we added a reference to support this methodology.
The experimental conditions (temperature, pH etc) seem to be identical with the acclimatisation ones. Therefore Line 146: the sentence should just acknowledge the identical conditions.
R2: We agree with the reviewer. The changes were made accordingly.
The results should try to explain or even acknowledge the high variability of data in the control samples, especially for SOD and Ubiquitin (Fig 2 and 3). Conclusions. Line 445: "The short exposure of adult...".
R2: The high variability of data in control can be attributed to sex or different genetic patrimony (fish are wild type). However, each control belonged to different lots of fish so it is also possible that this can influence some of the biomarkers analysed. Nonetheless, the variability between controls was not significantly different (p<0.05) meaning that they were not different.
Therefore, we added the following sentence to the discussion:
“Additionally, the high variability found in the controls for some biomarkers (e.g. SOD, Ubiquitin) can be attributed to factors such as individual variability, sex, genetic heritage or other biological and environmental factors to which the animal has been subjected [84]. Nonetheless, the statistics showed no significant differences between each batch of controls."
Line 445: "The short exposure of adult...".
R2: Changed as suggested.
The authors should discuss another factor, potentially responsible for the low or moderate effects found in this present study - the short exposure. I recommend Nesler et al, 2019 (Toxicology in vitro) - the potential of high impact after long exposure compared to low exposure is highlighted here.
R2: We agree with the reviewer. Comments on this issue were added to discussion as suggested by the reviewer including references. However, current scientific literature on this issue is often contradictory.
Interestingly, Brkic 2018 (http://dx.doi.org/10.5772/intechopen.71428) refers that “both short- and long-term safety of QDs will need to be established in toxicological studies in clinically relevant animal models”.
We added to the manuscript the following sentence:
"Furthermore, although some studies indicate low toxicity for short-term QDs exposure, other studies point increased toxicity when longer exposure periods are considered, as shown by Aye et al. [87], who studied the long-term in vitro toxicity effects of lipid-coated CdSe/ZnS QDs or Galeone et al. [88] who found long-term toxicity of CdSe-ZnS QDs in Drosophila melanogaster."
P.S. We were not able to find the suggested reference “Nesler 2019 at the Journal Toxicology in Vitro”? is it spelled correctly?
Round 2
Reviewer 1 Report
The authors address many of the points raised in the review. The authors need to further address and explain the continuing high concentration of the Zn in the 100 ug/L even when the quantum dots are removed from solution. None of the other Zn combinations have this high of a Zn concentration.
Authors should explain the analysis in the supplemental table. Is this a new preparation without fish added to the mixture? If the food is causing the increased Zn, the authors should prepare a solution with the quantum dots in fish medium for the same time as in the experiment, remove the quantum dots and measure the concentration of Zn in solution. A quantity of food should be tested to determine whether the high concentration of Zn can be attributed to the increases in recorded Zn.
The comment about unknown contamination suggests experimental error. Unknown contamination should not be an issue in these controlled experiments. The experimental error brings into question the results obtained when fish are exposed to the 100 up/L concentration of Zn. If this is experimental error the Zn exposures at 100ug/L should be redone.
Figure 3 b 100 ug/L is missing the g in the label.
Author Response
Reviewers' Comments to Authors and authors answers:
Reviewer 1
The authors address many of the points raised in the review. The authors need to further address and explain the continuing high concentration of the Zn in the 100 ug/L even when the quantum dots are removed from the solution. None of the other Zn combinations has this high of a Zn concentration.
R: According to an EU report, ENMs in water and under environmental conditions may remain intact (some portion) or suffer dissolution, speciation, biological or chemical transformation or react with water and CO2. They also aggregate or disaggregate and settle. The extent of this phenomena will depend on the type of ENMs and concentrations. Thus, when under realistic environmental conditions such as in aquarium waters the suspensions of nanomaterials are generally expected to be unstable: i.e. upon collision, particles may approach each other close enough for attractive Van der Waals forces to become dominant over repulsive electrostatic forces and steric hindrance. Consequently, particles adhere to each other and then settle due to gravity. Likewise, surface modification of nanomaterials can influence the environmental fate and behaviour, also depending on the nature of the ENMs. In fact, due to the interactions of nanomaterials with various components of the environmental system, generally, near-zero concentrations of the nanomaterial in its original form would be expected. (http://ec.europa.eu/health/ph_risk/documents/synth_report.pdf)
Therefore, for Zn at 100 µg/L, diverse factors may act (e.g dispersion, type and nature of nanomaterial, interaction, aggregation, oxidation, etc.) influencing the availability of Zn. In fact, different concentrations of nanomaterials may behave differently leading to different patterns of distribution within the water column. As explained above we would expect a great reduction of the element in its ionic form but again this will depend on many factors.
Therefore, after removing the QDs from the water samples, we attributed the high levels of Zn to a higher release rate of this element from the QD core. As mentioned by other authors, Zn dissociates more easily from its core than Cd. We think that some phenomena related with aggregation occurs at this intermediate concentration (100 µg/L) leading to a higher release of Zn ions and therefore to higher concentration in water although 30% below the nominal concentration as shown in table SM1.
That is, 10 µg/L is a very small concentration forming fewer aggregates than at 100 µg/L and less concentration means also less amounts of ions released. At 1000 µg/L concentrations are higher forming more larger aggregates (> 1µ when tested isolated and >596 nm when tested combined). On the other hand, at 1000 µg/L the larger aggregates sink and become less available on the water column when water samples are collected. As mentioned in the manuscript text larger aggregates tend to release fewer metal ions. Thus, at 100 µg/L, although aggregate formation is around 416 nm, some aggregates are still in the water column when sampling is performed which may lead to some overestimation of values.
When QDs are tested combined (different in its nature) the results should be analysed differently as various phenomena can act since both QDs may interact with each other causing different release rates as well as forming aggregates that differ in size and composition as referred in the previous EU report.
We rewrote some sentences in the manuscript (highlighted in red) for a better explanation of this subject.
Authors should explain the analysis in the supplemental table. Is this a new preparation without fish added to the mixture? If the food is causing the increased Zn, the authors should prepare a solution with the quantum dots in fish medium for the same time as in the experiment, remove the quantum dots and measure the concentration of Zn in solution. A quantity of food should be tested to determine whether the high concentration of Zn can be attributed to the increases in recorded Zn.
R: Yes. This was a new experiment performed after a previous suggestion from the reviewer to which we are grateful. It was performed only to evaluate the amount of both elements after the removal of both QDs by centrifugation. The assay conditions were similar as previous trials, but this time no fish and no food were added to the glass containers. Thus, the objective was to study the release of Zn and Cd ions from QDs core, without interference from food, urine and faeces. A better explanation of this new assay has been added to the M&M section, including sample treatment.
Consequently, the high levels that were still found in water added with nominal 100 µg Zn-QD/L cannot be a contribution of food or even fish. However, the amount of Zn determined in water samples after centrifugation is lower than the nominal concentration. Although levels may be high, they can be explained by the high dissociation of Zn from the core of QDs, as explained in the manuscript. In addition, at 100 µg/L the sinking of QDs aggregates is much lower than at 1000 µg/L and thus it is possible that more aggregates are present in the water column when the samples were collected.
The comment about unknown contamination suggests experimental error. Unknown contamination should not be an issue in these controlled experiments. The experimental error brings into question the results obtained when fish are exposed to the 100 ug/L concentration of Zn. If this is experimental error the Zn exposures at 100ug/L should be redone.
R: The comment on possible potential contamination in the manuscript was incorporated in a previous review to explain the result to the elevated levels of Zn above the nominal concentration. We strongly agree with the reviewer's opinion and it is not an adequate explanation for the observed total Zn levels. Therefore, we removed it from the manuscript.
Considering that in the assay that evaluated the release of ions from QDs core into the water (Table SM1), there was no fish or food and Zn levels remained elevated (about 70 µg/L, although below nominal concentration), we must conclude that this Zn did not come from contamination of food or fish.
We believe that this was not an experimental error, as shown by the results obtained in the experiment where the QDs were removed (table SM1) continuing to show high levels of Zn at 100 µg/L. After an extensive review of existing scientific literature, a possible explanation is related to samples collection, which may have led to an overestimation of Zn in the water samples. In fact, if upon sample collection a few more aggregates are collected then it may influence ICP-AES measurements overestimating the results.
On one hand, it is possible that at nominal concentrations of 100 µg/L when taking samples - they contained more aggregated QDs than at 10 µg/L (less concentrated). On the other hand, at 1000 µg/L the aggregates exceeded 1µ in size, showing a higher tendency to sink and therefore were not present at a high number in the water column from which the samples were taken. Thus, more aggregates present in heterogeneous samples (aggregates of various sizes) may have led to a bias in the ICP-AES results, overestimating the real value present in the water.
That is, as a result of a non-homogeneous dispersion of QDs among treatments, and in the specific case of the intermediate concentration (100 µg Zn-QDs/L), this resulted in the samples containing a greater number of aggregates per mL than at the other tested concentrations and thus biasing the results. Hence, the elevated value in the specific case of zinc is not due to an experimental error per se but to a bias caused by differences in concentrations and aggregation of QDs in the water.
Furthermore, the techniques commonly used for ENMs characterization when employed alone, show serious limitations when applied to complex samples. These constraints are overcome by using diverse methods to fully understand ENMs behaviour (e.g. DLS, TEM, XRD, XANES). In any case, the feasibility of the direct analysis of suspensions by ICP techniques, specially ICP-MS, depends on the composition and size of the particles proving that particulate and dissolved species behave in the plasma in a similar way. However, this behaviour cannot be extended to any nanoparticle (Laborda et al. 2016; http://dx.doi.org/10.1016/j.aca.2015.11.008).
In ICP-AES, if any aggregates are still present and were not completely digested then this will also influence the signal peak and thus ICP measurements which can lead to a bias on measurements. On the other hand, if some aggregates are collected in the water samples, then after digestion we will have higher levels of the element analysed. This occurs because nanomaterials do not dissolve in water and instead they form aggregates.
In ICP-AES, if any aggregate is still present and not fully digested, it will also influence the signal peak and therefore ICP measurements may lead to a bias in the measurements. On the other hand, if some aggregates are collected in the water samples after digestion, we will have potentially higher levels of the analyzed element. This is because nanomaterials do not dissolve in water and instead, we obtain a mixture of aggregates, released ions and dispersed nanomaterial. This forms a complex and sometimes difficult to analyze matrix.
As noted by the reviewer, this phenomenon was observed only for Zn and for this concentration. In the case of combined QDs, these should be discussed differently because in water the two types of QDs exhibit distinct behaviour from when isolated and tend to aggregate together and therefore any comparison with their behaviour in its isolated form must be done with caution.
Thus, more information was added to the discussion providing another explanation for the observed Zn values in the water.
However, due to the possible uncertainty of the result, because the water samples may have more aggregates of Zn then if the reviewer considers that this deserves further clarification, we can repeat the analysis of this element in water as several samples were taken during the test and stored frozen.
However, we were only able to do this between 6 and 15 January 2020, due to the closure of laboratories until 6 December due to this holiday period.
Finally, it must be noticed that according to an EU report the appropriate metrics of the measurement of manufactured nanomaterials in relation to environmental risk assessment is still under discussion. Meaning that there is still some uncertainty on this subject (https://ec.europa.eu/health/scientific_committees/opinions_layman/nanomaterials/en/l-3/6.htm#0; http://ec.europa.eu/health/ph_risk/committees/04_scenihr/docs/scenihr_o_003b.pdf).
In fact, the behaviour of ENMs in water, namely water containing organisms, still deserves further attention as many studies produce different results due to numerous physical, chemical and biological variables that influence the ENMs behaviour and availability.
Figure 3 b 100 ug/L is missing the g in the label.
R: the missing g was added to Fig.3b;
·We noticed an error on the scale of Fig. 3a (numbers (,) corrected to dot).
